# Fecal pollution can explain antibiotic resistance gene abundances in anthropogenically impacted environments

Antti Karkman [1,2,3], Katariina Pärnänen[4] & D.G.Joakim Larsson [1,2]

Discharge of treated sewage leads to release of antibiotic resistant bacteria, resistance genes and antibiotic residues to the environment. However, it is unclear whether increased abundance of antibiotic resistance genes in sewage and sewage-impacted environments is due to on-site selection pressure by residual antibiotics, or is simply a result of fecal contamination with resistant bacteria. Here we analyze relative resistance gene abundance and accompanying extent of fecal pollution in publicly available metagenomic data, using crAssphage sequences as a marker of human fecal contamination (crAssphage is a bacteriophage that is exceptionally abundant in, and specific to, human feces). We find that the presence of resistance genes can largely be explained by fecal pollution, with no clear signs of selection in the environment, with the exception of environments polluted by very high levels of antibiotics from manufacturing, where selection is evident. Our results demonstrate the necessity to take into account fecal pollution levels to avoid making erroneous assumptions regarding environmental selection of antibiotic resistance.

[1] Department of Infectious Diseases, Institute of Biomedicine, The Sahlgrenska Academy, University of Gothenburg, Guldhedsgatan 10, SE-413 46 Gothenburg, Sweden. [2] Center for Antibiotic Resistance research (CARe) at University of Gothenburg, P.O. Box 440SE-40530 Gothenburg, Sweden. [3] Faculty of Biological and Environmental Sciences, University of Helsinki, Helsinki 00014, Finland. [4] Department of Microbiology, University of Helsinki, Helsinki 00014, Finland. Correspondence and requests for materials should be addressed to A.K. (email: antti.karkman@helsinki.fi)

Growing concern over the threat posed by antibiotic resistant bacteria to human health has turned attention also to the environmental dimensions of the problem. Only fairly recently, the role of the environment as a source and dissemination route for antibiotic resistance has been acknowledged[1–3]. Treated effluent from wastewater treatment plants (WWTPs) is one of the most important point sources of resistant bacteria and resistance genes release to the environment[4,5]. Along with gut microbes, which contain a wide array of resistance determinants[6], antibiotics consumed by humans and animals are released into the environment in urine and fecal material contained in treated wastewaters and sludge applied to land.

Untreated sewage contains bacteria from human, animal, and environmental origin and a mixture of sub-therapeutic concentrations of antibiotics and other co-selective agents[4]. For these reasons WWTPs have been considered hotspots for antibiotic resistance emergence and dissemination[5,7]. While WWTPs are generally quite effective in removing antibiotic resistant bacteria (ARB) and resistance genes (ARGs) from the raw sewage[8–11] they are important point sources for ARBs and ARGs to the environment due to the large volumes released. The receiving environments form another possible hotspot for antibiotic resistance dissemination when bacteria originating from sewage and fecal material come in contact with environmental bacteria[2].

Although a distance-decay-effect can be found from point sources, there has been limited efforts to separate between enrichment of ARGs as a result of simple dissemination versus on-site selection[12,13]. It has been speculated that ARGs could be selected in the receiving environment by antibiotics and other co-selective agents originating from the WWTP, contributing to further dissemination of ARGs to environmental bacteria[3,14–16]. However, there are experimental studies (using complex microbial communities) suggesting that the concentrations of selective agents in a sewage-impacted environment might not be sufficient to cause selection[17,18] despite that competition experiments with two strains occasionally suggest otherwise (reviewed in ref. [19]). If there is selection for antibiotic resistance either in WWTPs or receiving environments, it could promote the emergence of novel resistance mechanisms and the dissemination of existing ARGs. A selection pressure would favor evolution of novel resistance mechanisms and the maintenance of recently transferred ARGs in the new host[2].

To determine the resistance levels in WWTPs and receiving environments several methods have been used including selective culturing of indicator bacteria, quantitative PCR (qPCR) of ARGs and metagenomics[8–11,20,21]. The class 1 integron (CL1) integrase gene has often been used as a proxy for anthropogenic impact and total ARG abundance with good resolution[22]. However, as CL1s can contain a wide array of ARGs and, thus, can be subjected to selection themselves, they are not an independent measure which can be used to distinguish between environmental selection or simple dissemination of ARGs from fecal matter. Fecal pollution levels have rarely been incorporated in the determination of possible selection or dissemination of ARGs. Detecting fecal marker bacteria using metagenomics is often difficult due to the low abundance of common marker bacteria in the whole community[23]. A robust marker for fecal pollution provides the means for distinguishing between on-site selection and dissemination of the genes versus accumulation and to observe decrease in ARGs which is due to dilution of fecal pollution in the receiving environments. If ARG abundance shows a strong and positive correlation with a fecal marker, it would indicate that the ARG abundance could be explained by fecal contamination alone without involving environmental selection as an explanation, while if ARG abundance is not well correlated with the fecal marker, it would suggest that other processes, like selection and/or HGT, is explaining elevated ARG abundances in addition to or instead of fecal contamination. Therefore, taking fecal pollution into account in models of ARB and ARG pollution from sewage would help in determining the critical control points for antibiotic resistance selection by enabling distinguishing between mere accumulation and selection.

Recently, crAssphage, a bacteriophage was identified from human fecal metagenomes[24]. The phage, which infects *Bacteroides intestinalis*[25], is highly abundant in the human intestine, as ~1.7% of human fecal metagenome reads align to it, and it is six times more abundant in public metagenomes compared to all the other known phages together[24] while being rare in feces from other animals[26]. It also performs equally well or better than traditional fecal markers in qPCR assays[27,28]. Given its high abundance, it can be used in metagenomic studies for estimating the human fecal pollution levels in the environment[29,30]. crAssphage has been observed to correlate with ARGs in storm drain outfalls[31] and thus, could be a good candidate for estimating the effect of fecal pollution in ARG dynamics. Moreover, it is not physically linked to resistance genes like integrons are and hence likely not co-selected with ARGs. Still, the environmental ecology of crAssphage is not fully understood and we do not know whether it can replicate also outside the gut environment. Another fecal *Bacteroides* phage, ϕB124-14, has been used for microbial source tracking but unlike crAssphage, it is also abundant in porcine and bovine gut[32], therefore making it possibly a good candidate for estimating the combined effect of human and production animal fecal pollution.

In this study, we analyze a selected set of public metagenomes from sewage-polluted environments and, in addition, nearly 500 metagenomes from MG-RAST. We show that in practically all the studied environments, the ARG, CL1, and MGE abundances correlate with fecal pollution levels with no evident signs of selection or dissemination of the resistance genes, except in sediments polluted with wastewater from drug manufacturing containing exceptionally high levels of antibiotics[33,34]. Using fecal samples from ref. [35] and the Integrative Human Microbiome project (iHMP)[36], we confirm the independence of mobile ARGs and the crAssphage in feces. The latter is crucial to be detect possible selection in environmental samples. Taken together, this approach can help in disentangling environmental on-site selection and/or horizontal dissemination of ARGs from passive dissemination/enrichment by fecal pollution.

## Results and Discussion

**Fecal pollution and ARG patterns in sewage polluted environments.** Concerns about the elevated levels of antibiotic resistant bacteria and resistance genes in the receiving environments of treated sewage have been raised in several publications[3,14,16,37,38]. However, relying solely on abundance data, it is difficult to determine whether the increase is explained by selection and dissemination of ARGs from bacteria originating from the WWTP to environmental microbiota or by the continuous input of fecal bacteria. To elucidate this question, we analyzed mobile antibiotic resistance genes in metagenomes from environments with anthropogenic impact from sewage discharge and correlated the total abundance of the mobile ARGs with a human fecal pollution marker, crAssphage, abundance. Our results show that the observed ARG abundances strongly correlate with crAssphage, meaning that the fecal pollution levels largely explain the observed abundances and there are no clear signs of wide scale selection or dissemination of antibiotic resistance in the affected environments.

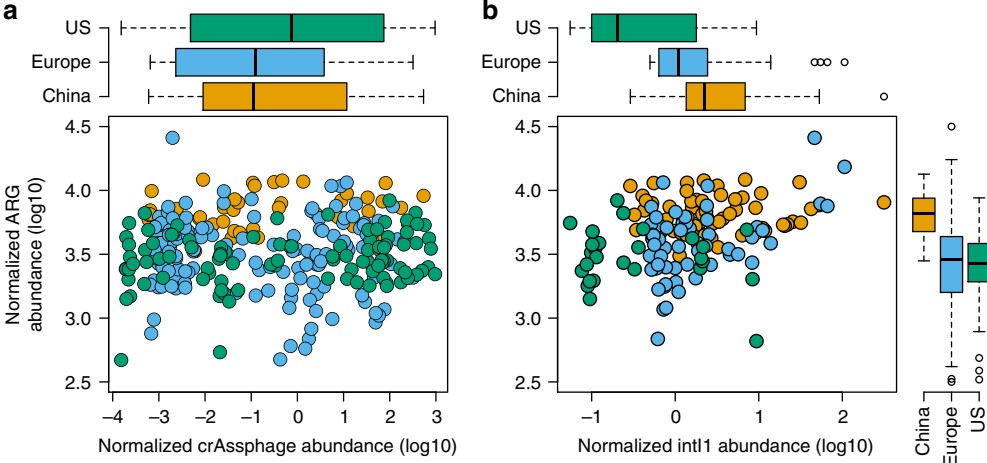

**Fig. 1** Abundance of antibiotic resistance genes, *intI1* gene and crAssphage in human fecal metagenomes. Mobile antibiotic resistance gene abundance in relation to crAssphage (**a**) and *intl1* gene (**b**) abundance in studied human fecal metagenomes from[35] and Integrative Human Microbiome Project (iHMP). We found no correlation between the total mobile ARG abundance and crAssphage abundance in human fecal metagenomes confirming the independence of the abundance of mobile ARGs from crAssphage abundance. In boxplots the lower hinge represents 25% quantile, upper hinge 75% quantile, and center line the median. Notches are calculated with the formula median ± 1.58 × interquartile range/sqrt (*n*)

**crAssphage and ARG dynamics in human fecal metagenomes**. To determine the independence of crAssphage and mobile ARGs in human feces, we analyzed fecal metagenomes from 74 Chinese, 234 European[35], and 141 subjects from the USA (HMP) (Supplementary Data 1) for ARG and crAssphage abundance. The screening of fecal metagenomes proved that crAssphage had a potential for revealing selection dynamics in receiving environments by correlating ARG abundance with the phage abundance in environments with human fecal pollution. We did not find any correlation between the ARG and crAssphage abundance in fecal metagenomes of the studied populations (linear regression, $F = 25.51$, adj. $R^2 = 0.21$, $p > 0.05$, Fig. 1a, Supplementary Data 2), confirming that ARG abundance is independent from crAssphage abundance. On the other hand, the correlation between total ARG abundance and *intI1* gene abundance was significant when taking in to account the different base levels of ARG abundance in the populations (linear regression, $F = 20.55$, adj. $R^2 = 0.29$, $p < 0.05$, Fig. 1b, Supplementary Data 2) showing the expected dependence between ARGs and CL1s.

We observed that the crAssphage abundance was similar in all cohorts, even though the abundance varied dramatically between individual subjects (Fig. 1a). The uniform abundance of crAssphage across the studied populations suggests that crAssphage could be used as a fecal pollution marker globally, which is in line with previous observations[39]. However, it should be noted that in another study, crAssphage was reported to be less abundant in sewage from Asia and Africa compared to Europe and US[30]. In our analysis, the total relative ARG abundances were similar in US and European subjects, while the Chinese subjects had higher relative abundance of resistance genes in their fecal metagenomes (Analysis of variance, Tukey *post-hoc* test, adj. $p < 0.05$ for both, Fig. 1b). The higher ARG levels in Chinese subjects compared to Europeans, using the same samples which were analyzed in here, has been reported earlier[35]. Differences on the ARG levels on population level might originate from the historical amount of antibiotics used regionally as well as the degree of transmission control. However, even with differences on the initial crAssphage to ARG ratio in different populations, deviations from this relationship in the receiving environment will reveal possible selection hotspots.

**Industrially polluted sediment is a hotspot for ARG selection**. To determine the relationship between ARG abundance and fecal pollution in environments with anthropogenic impact, we selected metagenomic studies from literature where sequence data was available (Supplementary Data 1). In all environments, except in environments polluted directly by wastewater from the manufacturing of antibiotics (see below), the total ARG abundance positively correlated with the crAssphage abundance, showing that in these environments the ARG abundance could largely be explained by the extent of fecal pollution (linear regression, $F = 32.51$, adj. $R^2 = 0.75$, $p < 0.05$, Fig. 2, Supplementary Data 2). The highest ARG abundances were detected in industrially polluted Indian sediments and in hospital and WWTP effluents from UK and Singapore. Lower levels of ARGs were detected in sediments and river water downstream of a hospital as well as WWTP effluent discharge points, in line with the diluted fecal material quantified with crAssphage abundance (Supplementary Data 2).

Antibiotic resistance genes conferring resistance to different classes of antibiotics followed the same trend in all datasets (Supplementary Figure 1, Supplementary Data 4), with the exception for quinolones in two datasets. The deviations, however, were explained by a single gene, *oqxB*, coding for part of an RND efflux pump with very broad substrate specificity[40]. Also, the counterpart of the mobilized RND system, *oqxA* did not follow the pattern of *oqxB*. This suggests that the elevated presence of reads matching *oqxB* in some samples reflects a taxonomic change towards bacteria carrying the gene chromosomally, and not likely as a consequence of quinolone exposure and selection by quinolones.

Overall, the strong and consistent correlation between crAssphage and ARGs (also across antibiotic classes) suggests the observed enriched ARG abundance is primarily due to fecal contamination rather than selection of the ARGs in the downstream environments. Similarly, the ARG richness, total abundance of MGEs and *intI1* gene abundance could also be explained by fecal pollution levels (Supplementary Figure 2).

Indian sediments polluted with exceptionally high levels of antibiotics from drug manufacturing[33,41] provided a clear exception to the general trend. Here, ARG levels were very high, at the same time crAssphage was not detected in most of these

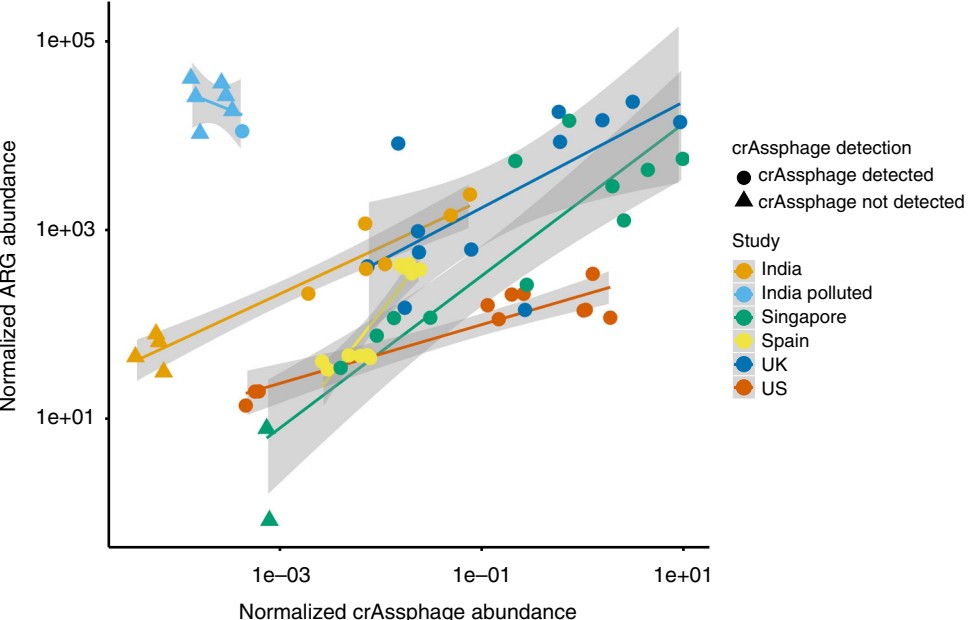

**Fig. 2** Correlation between ARG abundance and crAssphage abundance in environments with pollution from WWTPs, hospitals, or drug manufacturing. The environments effected by drug manufacturing are polluted with exceptionally high levels of antibiotics, and the analyses show clear selection for antibiotic resistance as the ARG abundance cannot be explained by fecal pollution. ARG abundance and crAssphage abundance were normalized with total nucleotide count in the metagenomes. For samples where we did not detect crAssphage (indicated by triangles), half of the detection limit (corresponding to one read mapping to crAssphage) was used and normalized to the total nucleotide count

samples indicating that they contained very little human fecal material. These and other sediments from the same industrially polluted area contain the highest levels of antibiotics ever measured in the environment[33]. The exceptionally high abundance of ARGs and together with very few or no crAssphage reads observed in the metagenomes provides strong support for the hypothesis that direct selection for resistance rather than fecal contamination explains the abundance of ARGs in the polluted sediments. It should be noted that the DNA in these samples was amplified using RepliG prior to sequencing, which could favor small circular plasmids and thereby inflate the counts for genes carried by these[34]. However, even by removing the most abundant ARGs, the pattern was still consistent (Supplementary Figure 3).

**ARG abundance is largely explained by fecal matter, not selection.** To expand our analysis beyond the selected studies of polluted environments, we analyzed 484 publicly available metagenomes from MG-RAST, previously analyzed for ARGs in Pal et al. (2016) including environments with human fecal pollution. We were able to detect crAssphage only in samples taken from the Mississippi river (USA), WWTPs, activated sludge with high ammonia content, Beijing air and from laboratory mice. Fecal pollution correlated with the observed ARG abundance in the river water, Beijing air and in wastewaters (excluding activated sludge with high ammonia content) (Fig. 3). The strong correlation of ARGs and crAssphage in the large-scale analysis of metagenomes with varying levels of fecal contamination is in line with our results in the selected anthropogenically polluted sites, confirming that also in these environments the mobile ARG abundance could be explained with fecal pollution and no apparent signs of large-scale selection or horizontal gene dissemination could be detected (Fig. 3, Supplementary Data 2). Interestingly, this was true also for Beijing smog samples, hosting

a particular high diversity of ARGs[42]. The activated sludge samples with high ammonia concentrations were from a laboratory experiment[43] and conclusions about possible selection during wastewater treatment in these samples should not be drawn.

A sometimes high abundance of crAssphage and consistently high abundances of ARGs were found from mice gut metagenomes in MG-RAST. All of these mice were from murine model experiments and were not given antibiotics[44–46]. It is known that laboratory mice gut microbiota bear some similarities to human gut microbiota[47] and differs from the gut microbiota of wild mice[48,49], which could explain these findings. As in human feces, the crAssphage abundance in mice feces was not linked to the total ARG abundance.

The samples with elevated resistance levels where we could not detect crAssphage were either animal fecal metagenomes or environments polluted with animal feces (Supplementary Figure 4). Since crAssphage is reported to be rather specific to human feces, we accordingly could not find a correlation with ARG abundance in these environments. To further verify the lower abundance of crAssphage in other animals compared to humans we analyzed 12 chicken gut[50], 100 pig gut[51], and 42 cow rumen[52] metagenomes and did not detect crAssphage, in agreement with the results of others[24,26,29,30]. Hence, crAssphage is very specific for detecting anthropogenic pollution. To disentangle selection from fecal pollution in environments polluted with animal feces, additional markers for animal feces would be needed. We therefore determined the abundance of φB124-14 phage in the MG-RAST metagenomes. We found it to be less abundant than crAssphage in human impacted environments and not to perform well with other pollution sources such as animal feces either. Overall φB124-14 was detected in fewer metagenomes than crAssphage (Supplementary Data 3). The lower abundance of φB124-14 in the studied metagenomes suggests it is not suitable to be used for tracking fecal pollution in metagenomic data sets. However, a qPCR-based approach with

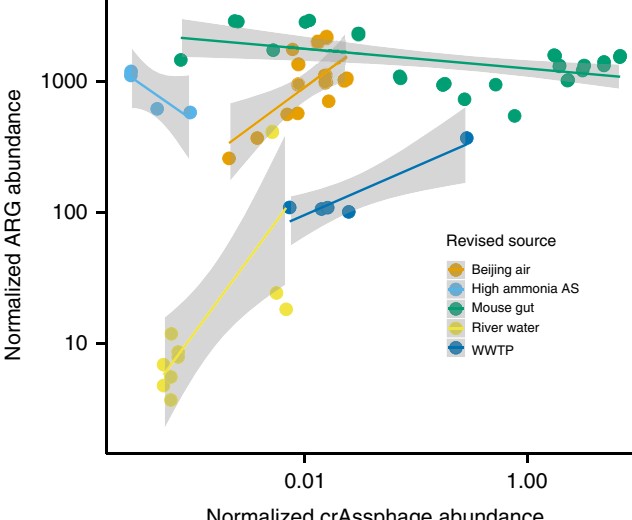

**Fig. 3** The correlation between crAssphage abundance and total ARG abundance in MG-RAST metagenomes where crAssphage was detected. Only in the mice gut and sludge with high ammonia concentration we could not see a correlation (discussed in main text). Original MG-RAST feature annotations were revised manually using the project and sample descriptions due to the misleading information in original annotations. Original annotations can be seen in Supplementary Data 1

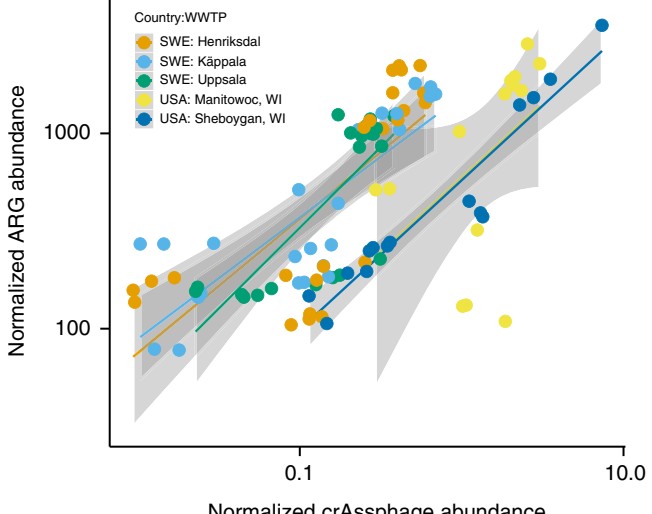

**Fig. 4** ARG and crAssphage abundance in two US and three Swedish wastewater treatment plants showing similar correlation with different base level of resistance. Smoothing curves based on linear model separately for each plant are shown in gray with 95% confidence intervals

lower detection limit could give better results. Due to the low detection frequency, we did not use φB124-14 in this study to detect possible selection hotspots.

**Predicting antibiotic resistance gene abundance with crAssphage.** From the abundance of crAssphage, we were able to predict antibiotic resistance gene abundances with good accuracy (linear regression, $F = 34.76$, adj. $R^2 = 0.54$, $p < 0.05$) in the environments with anthropogenic impact in MG-RAST using a linear model constructed from the selected metagenomes from polluted environments (Beijing air, river water, & WWTP) (Supplementary Figure 5). However, due to the different base levels of resistance genes in the point source (sewage), the baseline (intercept) of ARG abundance cannot be predicted with high accuracy. It has been shown that different parts of the world have different resistance burden[35], which is seen also in fecal samples analyzed in this study (Fig. 1). Furthermore, different environments might have different levels of fecal pollution from domestic animals, which could explain some of the varying background levels of resistance when compared to the crAssphage abundance. However, when the baseline level of an environment is estimated, the model using crAssphage performs well in predicting the ARG abundance.

**Antibiotic resistance gene dynamics in wastewater treatment plants.** As noted earlier, WWTPs could potentially serve as hotspots for antibiotic resistance selection and horizontal gene transfer. To determine correlation between ARG abundance and fecal pollution in WWTPs during the treatment process, we analyzed metagenomes from three Swedish WWTPs[8] and two WWTPs in Wisconsin, USA[53]. These studies were selected since the sequence data was publicly available and accompanied by comprehensive metadata. The results show clearly the ARG abundance decreases from raw sewage to the treated sewage in parallel with a similar decrease in fecal material during the treatment process (Fig. 4). In sludge, the ARG abundance seems to be even lower than expected based on the crAssphage

abundance (Supplementary Figure 6). These results speak against WWTPs being hotspots with a strong antibiotic resistance selection acting as a driver for horizontal gene transfer[5,7] and show that at least in the studied treatment plants the treatment process eliminates ARGs along with fecal pollution with high efficiency. Although differential survival and growth of bacterial taxa during the treatment process carrying ARGs or crAssphage can add to noise in the analysis, we could see clear correlation between fecal pollution and ARG abundance along the treatment process indicating that there is no evident selection for resistance. The same was concluded in the Swedish WWTP study even though the authors found tetracycline and ciprofloxacin concentrations slightly above predicted selective concentrations in the influents[8]. A very recent culture-based study of over 4000 E. coli isolates from influent and effluent from Scandinavias's largest WWTP also found no support for within-species selection in the treatment plant[54]. Yet other studies have shown a reduction of ARGs during the treatment process[9,11]. Our results connect the reduction of ARGs to elimination of fecal material from sewage during the process. In all of the five treatment plants studied here, the correlation between crAssphage and ARG abundance was similar (Fig. 4). A linear model between ARG abundance and crAssphage detection confirmed that the correlation was significant and similar in both countries and all WWTPs (linear regression, $F = 71.26$, adj. $R^2 = 0.59$, $p < 0.05$). Only the intercepts differed between the WWTPs in Sweden and USA indicating that the proportion of ARGs to crAssphage in the sewage entering the plant was different in the two countries, but the ARG removal efficiencies of the treatment processes were on par for all treatment plants.

One would perhaps expect a similar ARG to crAssphage ratio in Swedish and US sewage influents, given the overall comparable resistance gene abundance in gut metagenomes of Swedish and US subjects[55]. Explanations behind the differences found might be technical and come from sample handling or be related to the sources of sewage entering the plant but unfortunately, we cannot asses that with the data available. The ARGs per metagenomic read were, on average, slightly higher and crAssphage more common in the US populations (refer to Fig. 1 to see differences in ARG abundances in different countries). This suggest that one should probably restrict comparisons of ARG to crAssphage

ratios to samples that are contaminated by the same population of people and prepared using similar protocols, as we have done in this study.

**Estimated resistance risk correlates with fecal pollution**. A computational pipeline estimating the potential of ARGs being associated with MGEs and mobilized to pathogens (resistome risk) was published recently[56]. We analyzed the same samples that were used for benchmarking the pipeline and found that the resistance risks as estimated by the authors[56] followed fecal pollution levels in each environment having human fecal pollution. The correlation was even stronger between the resistance risk and the total count of mobile ARGs than between resistance risk and fecal pollution in all environments (Supplementary Figure 7). Our analysis show that in one of the hospital samples there was elevated levels of ARGs compared to the fecal content (Supplementary Figure 7). However, their overall contribution to the resistance load at WWTPs has been shown to be small[57]. So, to fine-tune the risk assessment, we would propose to include the fecal content in to the analysis to be able to determine possible selection or dissemination scenarios (risks for evolution of resistance) with better precision, as opposed to situations where there mainly is a risk for transmission of already resistant pathogens[58].

Estimating the risk associated with environmental antibiotic resistance is far from simple. Besides the total amount of ARGs, it clear that association of resistance genes with MGEs and pathogens elevates the risk caused to human health[59]. However, detecting a resistance gene in the environment does not necessarily mean a risk for human health. Certainly, there is a higher probability of transfer when there are more transferrable genes and recipient cells together and when the genes are already on mobile genetic elements ready to be transferred to pathogens. However, selection plays a critical role in the processes required for the transferred gene to persist in the new host or a newly emerged gene to disseminate[60]. Using proxies for fecal pollution such as crAssphage enables detecting possible selection hotspots where the detected ARG abundances cannot be solely explained with human fecal contamination and thus, can help in assessing risks associated with elevated levels of ARGs in the environment.

In conclusion, our results provide a framework to help disentangling dissemination of resistant human fecal bacteria from the possible selection and horizontal gene transfer of resistance genes in the environment. We were able to detect true hotspots for antibiotic resistance gene selection in sediments receiving exceptionally high levels of antibiotics from industry. In addition, we show that in all other studied environments receiving anthropogenic waste, there was no clear evidence of wide scale selection. On the contrary, the ARG abundance correlated strongly with fecal pollution, which does not support the prevailing speculations that major selection for antibiotic resistance occurs in WWTPs or effluent receiving environments. It should be noted that antibiotic concentrations were not measured in most studies. Knowing the concentrations of antibiotics in these environments would have provided an even deeper insight on the selection dynamics. A lack of apparent selection in these environments would mean that the emergence of new resistance determinants is less likely and the transferred resistance genes are not likely to be fixated on new hosts due to selection pressure. In the heavily polluted Indian sediments new resistance determinants are possibly more persistent and disseminate more efficiently in the population due to the competitive advantage they give to the receiving strains.

It should be noted that selection occurring on small scale, which does not affect the entire population, or selection of limited types of resistance genes would probably not be detected using this method. Furthermore, rare horizontal transfer events will definitely be missed and more precise methods are needed. Reported increases in relative proportion of individual ARGs in certain environments exposed to sewage/low levels of antibiotics could reflect such small-scale selection. However, community-based analyses, including qPCR or metagenomics, do not provide evidence for within-species selection. Hence, increases of individual ARGs would more easily be explained by simple changes in species composition, caused by other factors than the exposure to antibiotics. A lack of consistency with regards to the type of antibiotics the enriched ARGs provide resistance to, provides further support to the latter explanation[8].

Another important source of antibiotic resistance determinants is animal husbandry, which uses more antibiotics than are prescribed to humans. In the US animal manure exceeds the human sewage sludge by ~100-fold[61]. As crAssphage is abundant only in human fecal material it cannot be used to estimate fecal content of environments receiving animal manure and feces. However, it is likely that similar phages that are specific to different production animals could be identified and used in a similar fashion. Discovering more markers for fecal pollution would advance the field of studying antibiotic resistance in the environment. We argue that the use of crAssphage, or similar fecal pollution markers, should be incorporated broadly in studies determining the persistence, fate and possible selection of resistance genes originating from fecal pollution, which is likely the main source of resistance genes of environmental ARG pollution.

In terms of ranking risk in resistomes, we argue that the total count of ARGs or their genetic context does not give the whole picture of the actual risks. Without a way to estimate the extent of selection pressure in the environment, the total count of ARGs or the association of an ARG to mobile element adds only a few pieces to the entire puzzle of disentangling the risks related to resistance genes, especially when in most cases the genes are eventually diluted to near zero concentrations following the diminishing fecal pollution. Real-time PCR primers for detecting crAssphage are already available[27,28], so the use of metagenomics is not necessary, making the analysis quick, easy and affordable. When collecting data, we encountered many studies where sequencing data was not available, even though sometimes presented as available for download in public repositories. This is unfortunate, as all this data would have been valuable in this study. Re-analyzing these samples using markers for fecal pollution would expand our knowledge on resistance gene dynamics is diverse environments. To conclude, we argue that including a proxy for fecal pollution in future studies would enable a more comprehensive understanding of the antibiotic resistance dynamics in the receiving environments, providing more reliable estimates of the risk scenarios and perhaps most importantly, discovering true environmental hotspots for selection and dissemination of antibiotic resistance.

## Methods

**Data collection**. The crAssphage (NC_024711.1) and φB124-14 (HE608841.1) genomes were downloaded from GenBank and indexed for mapping using bowtie2-build[62].

Metagenomic studies on human impacted environments were searched from the literature and six studies where the sequencing data was available were selected and downloaded from either SRA or ENA[13,33,34,38,63–66]. The studies included samples from river and lake sediments, WWTP and hospital effluents and river water. Two wastewater treatment plant studies with comprehensive metadata were used to determine the impact of wastewater treatment on the ARG abundance and fecal content[8,53]. Accession numbers for all metagenomic data can be found from Supplementary Data 1. Many data sets from peer-reviewed studies that were candidates for being included in this study were unfortunately not made publicly available.

A set of 484 metagenomes from MG-RAST, excluding metagenomes from different human body sites, analyzed in[42], were used for the study. The metagenomes included soil, freshwater, marine, animal, wastewater, agricultural, and air samples. Full list of accessions and annotations are available in Supplementary Data 1.

To study the abundance of crAssphage and its association to the total ARG abundance in human gut metagenome samples, 74 Chinese and 234 European subjects were used from a previous study[35]. In addition, gut metagenomes from 141 US subjects were downloaded from the HMP portal (https://portal.hmpdacc.org/). Accessions and download links for HMP subjects can be found from Supplementary Data 1.

To determine the abundance of crAssphage in animal gut metagenomes, 12 chicken metagenomes[50], 100 pig metagenomes (a subset from a larger study)[51], and 42 cow rumen metagenomes[52] were downloaded from SRA. Accession numbers are given in Supplementary Data 1.

**crAssphage and ARG annotation**. Metagenomic reads were mapped against the crAssphage genome using bowtie2[62] and the crAssphage genome coverage was calculated using Samtools[67]. Only read pairs mapping in proper pairs were calculated in case of paired-end sequencing. For single end metagenomes, the mapped reads were filtered with quality value 10 using Samtools. The genome coverage was used as a measure for crAssphage abundance in the sample. ResFinder, a database of mobile, acquired antibiotic resistance genes[68] was translated from nucleotide sequences into amino acid sequences using Biopython. The longest ORF for each gene was selected to represent the gene in the database as some genes included upstream and/or downstream regions. MGE database[69] (available from: https://github.com/KatariinaParnanen/MobileGeneticElementDatabase) was used for MGE annotation by first removing all plasmid marker sequences. All remaining entries were translated to amino acid sequence as described earlier. ARGs and MGEs were annotated against the translated ResFinder/MGE database using DIAMOND blastx[70] with the following parameters: minimum identity 90%, minimum match length 20 AA. The parameters were chosen to be less conservative in order to identify also genes not stretching over the whole read. It should be noted that this approach can result in false-positives. In case of paired-end sequencing, matches on the second read were counted only if there was no match on the first read. Both crAssphage abundance and antibiotic resistance gene abundance were normalized with the total base pair count in the metagenome. Mapping against φB124-14 was done exactly as with crAssphage.

**Statistical analyses**. Both normalized crAssphage abundance and normalized total ARG abundance were log10 transformed and linear regression was done using the *lm* function in R v.3.2[71]. The ARG abundances in the MG-RAST metagenomes were predicted from the crAssphage abundance based on the linear model based on the selected studies using the function *predict* in R. Figures were drawn in R using the base graphic package and ggplot2 package v.2.2.1[72]. Smoothing curves using a linear model were drawn with function *geom_smooth* in ggplot2. Statistical comparison of mobile antibiotic resistance gene abundance between European, Chinese and US subjects was done using the *aov* function in R. A post-hoc test was done using the *TukeyHSD* function in R.

**Code availability**. All scripts and R code for the data analysis and statistics can be found at https://github.com/karkman/crassphage_project.

## Data availability

The sequence data analyzed in this study are available in public repositories with the accession codes given in Supplementary Data 1.

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

## Acknowledgements

This research was funded by the Swedish Research Councils VR (2015-02492) and FORMAS (942-2015-750) to D.G.J.L. and the Adlerbert Research Foundation to A.K. The Centre for Antibiotic Resistance Research at University of Gothenburg (www.care.gu.se) also supported the project.

## Author contributions

A.K. and D.G.J.L. designed the study. A.K. did all the bioinformatic analyses. A.K., D.G.J.L., and K.P. interpreted the results. A.K. drafted the manuscript with input from K.P. and D.G.J.L. All authors contributed to manuscript revisions and have read and approved the final version of the manuscript.
