## [Peer Review File · Nature Communications]

Reviewers' comments:

Reviewer #1 (Remarks to the Author):

The manuscript entitled, "Fecal pollution explains antibiotic resistance gene abundances in anthropogenically impacted environments" suggests that an observed association between the abundance of antibiotic resistance genes and human-associated virus genome in sewage can be used to reveal ARG dissemination mechanisms in wastewater treatment and environmental settings. Evidence is presented using previously published metagenomic DNA data sets combined with bioinformatic approaches. Authors conclude that trends in the abundance of resistant, human fecal bacteria do not support a transmission mechanism based on HGT in sewage treatment and surface water environs. Instead, findings suggest an alternative explanation where ARG occurrence is simply a predictable function of the corresponding human fecal pollution levels.

While the notion that there are predictable patterns between ARB occurrence and specific pollution sources of fecal waste is not new, the application of metagenomic DNA sequence data analysis approach using the recently described CrAssphage genome to elucidate patterns of ARG transmission in complex systems (i.e. sewage treatment and environmental samples) is. ARG risk assessment is incredibly difficult to study in environmental settings and is rapidly commanding the attention of public health officials across the world. This work represents a novel, useful, and important incremental step towards addressing this issue. Authors do a good job of providing a realistic interpretation of findings thoroughly discussing limitations, as well as outlining current research gaps.

Overall, the manuscript is well-written, organized, and clear. Reviewer recommends publication.

Reviewer #2 (Remarks to the Author):

- The paper deals with the drivers of increased AMR prevalence in sewage impacted environments. The authors claim to be able to disentangle selection from dissemination (they erroneously use transmission in the abstract which should be restricted for transmission to humans or animals).
- My initial response would be to suggest that studying associations between a human faecal phage and AMR genes in metagenomes, where phage abundance is used as a proxy for faecal pollution, does not disentangle selection from dissemination in terms of selection within the waste water system or even the environment as antibiotic residues are also likely to be associated with sewage load meaning that is still not possible to disentangle the two phenomena.
- It will of course be different in manufacturing waste polluted sites because the antibiotics enter the environment independently of faeces, which is not the case with municipal waste.
- There is also the possibility that selection for faecal taxa results in further phage proliferation, further reducing the utility of the approach to disentangle dissemination from selection.

Introduction:

- "However, as CL1s can contain a wide array of resistance genes and, thus, can be subjected to selection themselves, they are not an independent measure which can be used to assess the selection or dissemination of ARGs".

I'm not sure why the authors focus on the independence of CL1s from ARGs as a negative, it is exactly this linkage which makes them a useful proxy for a wide range of AMR mechanisms.

- “Moreover, it is not physically linked to resistance genes like integrons are and hence likely independent from ARGs and thus, could enable the determination of ARG horizontal gene transfer and selection patterns in receiving environments”.

The authors need to clarify how this would work as its not clear.

- “A robust marker for fecal pollution would provide the means for distinguishing between onsite selection and dissemination of the genes versus accumulation and to observe decrease in ARGs which is due to dilution of fecal pollution in the receiving environments”.

I don't follow this logic, due to the factors mentioned above, ie. co-correlation of faecal load and antibiotic concentration and the potential for further phage proliferation I'm not sure I accept the hypothesis. Because WWTP discharges are continuous and there will be a distance decay / dilution effect for ARGs and antibiotics one would expect there to be a correlation between ARGs and faecal pollution regardless of whether dissemination / or selection was the primary driver. The authors later say there is no association between the phage and ARG abundance in human feces which rings alarm bells.

- There also appears to be some uncertainty regarding the host of the crAssophage, so there is a possibility that this can replicate in environmental strains of Bacteroides and/or other taxa.

- Only 6 studies appear to be from human impacted environments, whereas there are 484 from environmental metagenomes (presumably not human impacted) and 308 metagenomes were from the human gut. Presumably the differences found may be a result of this significant disparity in sample size.

Methods

- If there is no correlation between the phage and ARGs in human feces why would it be a suitable marker and demonstrate a strong association with ARGs in effluents and receiving waters if it is just a matter of dilution during dissemination?

Results and discussion

- “Our results show that the observed ARG abundances strongly correlate with crAssophage, meaning that the fecal pollution levels largely explain the observed abundances and there are now clear signs of wide scale selection or dissemination of antibiotic resistance in the affected environments”.

Again the meaning is not clear hear, do the authors mean “no clear signs”? There is a growing body of literature demonstrating increased prevalence of ARGs in natural environments impacted with WWTP effluent / sewage sludge, so again I don't know what the authors mean or how they can conclude this from the study of a very small number of human impacted sample sites?

- “We did not find any correlation between the ARG and crAssophage abundance in fecal metagenomes of the studied populations (linear regression, $F = 25.51$, adj. $R^2 = 0.21$, $p > 0.05$, Figure 1, Suppl. Table 2), confirming that ARG abundance is independent from crAssophage abundance. On the other hand, the correlation between total ARG abundance and intI1 gene abundance was significant when taking in to account the different base levels of ARG abundance in the populations (linear regression, $F = 20.55$, adj. $R^2 = 0.29$, $p < 0.05$, Figure 1, Suppl Table 2) showing the expected dependence between ARGs and CL1s”.

Surely this undermines the suggestion in the introduction that 1) the phage is a good indicator of fecal pollution and 2) that CL1s are not good markers of AMR?!

- “In sludge, the ARG abundance seems to be even lower than expected based on the crAssphage abundance (Suppl. Figure 6). These results speak against WWTPs being hotspots with a strong antibiotic resistance selection acting as a driver for horizontal gene transfer (Guo et al., 2017; Rizzo et al., 2013) and show that at least in the studied treatment plants the treatment process eliminates ARGs along with fecal pollution with high efficiency”.

Perhaps because crAssphage are replicating in WWTPs or because the majority of sludge is made up of bacteria that replicate within WWTPs.

- “It should be noted that selection occurring on small scale, which does not affect the entire population, or selection of limited types of resistance genes would probably not be detected using this method. Furthermore, rare horizontal transfer events will definitely be missed and more precise methods are needed”.

This is probably what actually happens, if all ARGs increased significantly in prevalence in all taxa selection would be easy to detect.

In summary I don't think the manuscript is well argued and there are several logical problems with the argument and conclusions. There are also methodological issues based on the knowledge of the ecology of the specific phage and the small number of impacted metagenomes studied.

Reviewer #3 (Remarks to the Author):

This manuscript is of considerable interest as it concludes that the presence of antibiotic resistance genes in natural environments broadly correlates with the presence of human faecal pollution. If correct, this conclusion will have an important impact on our understanding of the spread of antibiotic resistance genes across the planet and the role of human activity in this process. The authors use as a marker for human faeces the recently discovered crAssphage which is claimed to be specific to human faeces. The specificity of this marker is cause for some concern: Garcia-Aljaro et al 2017 conclude that ‘crAssphage was also found in samples contaminated with faecal remains from certain animals, though in lower concentrations [approximately 100-fold] than in the human sources’ so it is not uniquely specific to humans. The authors have addressed this by analysing the abundance of another Bacteroides phage Φ B124-14, which is found in both humans and animals. However, this analysis is not shown, even though it is crucially important. It needs to be included and discussed in more detail in a revised version of this manuscript. As Suppl Fig 4 shows high levels of resistance genes can be present in samples from environments that are (presumably) contaminated with animal faeces. It would be very important to know the abundance of Φ B124-14 as a human + animal faecal marker in these samples. It appears that the authors may be overly focused on crAssphage and thus are too anthropocentric in their focus on human faecal pollution as animal faeces appear to be another major source of ARGs in the environment.

One major methodological issue is that there may be a real risk for spurious hits with a minimum match length of only 20 amino acids. Would it not be better to use as a cut-off a match length based on the length of the gene/protein (e.g. >90% of full-length gene/protein) and would this analysis importantly change the outcomes of the analyses? The somewhat unexpected observation that the abundance of oqxA and oqxB is poorly correlated, even though these genes are generally co-localised, may potentially be explained by ‘hitting’ a remotely related gene to either oqxA or oqxB that is identical over a stretch of 20 amino acids. I would like to see some analyses with a more stringent cut-off to determine whether this could really be an issue.

Minor issues:

- Please add page and line numbers. For any comment below I have simply copied the relevant line, which you should be able to find in the document using the ‘Find’ function.

- There a small number of minor issues with language or style in the manuscript. I have not identified all of them here.

Abstract

- 'This was possible by analyzing the abundance of a newly discovered phage': please name the phage in the abstract

Introduction

- 'Fecal pollution levels have rarely been incorporated in the determination of possible section': replace section by selection.

- The authors may want to add a reference to the recently published study by Stachler et al (Environ. Sci. Technol., 2018, 52 (13), pp 7505–7512) to the manuscript.

Materials and methods

- I initially read '12 chicken (Xiong et al., 2018), a subset of 100 samples from a bigger pig metagenome study (Xiao et al., 2016)' as meaning that the 12 chicken samples were extracted from a larger pig metagenome study that was not further analysed. Perhaps this needs to be rephrased for clarity.

- It is unclear what the authors mean with 'longest ORF was selected for each gene'. As ResFinder holds unique genes it is not entirely clear what the authors have done. E.g. have they clustered genes to a certain threshold of nucleotide conservation and then selected as a representative gene for the cluster the longest gene in each 'gene cluster'?

Results and discussion

- 'and there are now clear signs of wide scale': this is a crucial typo as I believe the authors mean 'no clear signs'

- More information on the types of resistance genes should be provided, e.g. by providing a supplementary dataset that lists the abundance of each individual resistance gene. This will reveal whether there are specific antibiotic resistance genes that are highly abundant and that mostly determine the 'normalized ARG abundance' as used in the main text.

- 'Using proxies for fecal pollution such as crAssphage enable detecting selection and thus, can help in assessing risks associated with it.' This is clearly wrong: crAssphage may be used to detect the presence of human faeces and the resistant bacteria that are associated with it, but it does not detect selection for antibiotic resistance in an environment.

Supplementary data

Supplementary tables 1 and 2 need informative legends.

Figures

Is the y-axis in Fig 2 – 4 correct? E.g the normalised ARG abundance cannot be 10^{1000} but should probably be 10^3

Below our point-by-point response to the referees' comments.

Reviewer #1 (Remarks to the Author):

The manuscript entitled, "Fecal pollution explains antibiotic resistance gene abundances in anthropogenically impacted environments" suggests that an observed association between the abundance of antibiotic resistance genes and human-associated virus genome in sewage can be used to reveal ARG dissemination mechanisms in wastewater treatment and environmental settings. Evidence is presented using previously published metagenomic DNA data sets combined with bioinformatic approaches. Authors conclude that trends in the abundance of resistant, human fecal bacteria do not support a transmission mechanism based on HGT in sewage treatment and surface water environs. Instead, findings suggest an alternative explanation where ARG occurrence is simply a predictable function of the corresponding human fecal pollution levels.

While the notion that there are predictable patterns between ARB occurrence and specific pollution sources of fecal waste is not new, the application of metagenomic DNA sequence data analysis approach using the recently described CrAssphage genome to elucidate patterns of ARG transmission in complex systems (i.e. sewage treatment and environmental samples) is. ARG risk assessment is incredibly difficult to study in environmental settings and is rapidly commanding the attention of public health officials across the world. This work represents a novel, useful, and important incremental step towards addressing this issue. Authors do a good job of providing a realistic interpretation of findings thoroughly discussing limitations, as well as outlining current research gaps.

Overall, the manuscript is well-written, organized, and clear. Reviewer recommends publication.

We thank reviewer #1 for acknowledging the novelty and importance of our work and for finding our discussion and interpretation well balanced.

Reviewer #2 (Remarks to the Author):

- The paper deals with the drivers of increased AMR prevalence in sewage impacted environments. The authors claim to be able to disentangle selection from dissemination (they erroneously use transmission in the abstract which should be restricted for transmission to humans or animals).

We agree that we should have used the term dissemination throughout, not transmission. This is now corrected.

- My initial response would be to suggest that studying associations between a human faecal phage and AMR genes in metagenomes, where phage abundance is used as a proxy for faecal pollution, does not disentangle selection from dissemination in terms of selection within the waste water system or even the environment as antibiotic residues are also likely to be associated with sewage load meaning that is still not possible to disentangle the two phenomena.

Yes, both the presence of ARGs and antibiotic load, as the reviewer point out, is indeed likely to be associated with sewage load. Such a correlation would not require any selection by the antibiotics within the wastewater system or the environment, since both ARGs and antibiotics come from the same source (human excretions). The less sewage, the less ARGs and antibiotics and phage. This is in fact the major point we make in the paper. Based on the above comment and some of the comments below, we have reasons to believe that we have not succeeded in making the concept clear enough. We have therefore tried to be a bit more explicit in the introduction on the conceptual parts, i.e. that a stable ratio between a marker for fecal pollution and ARGs indicates that fecal dissemination indeed is sufficient for explaining the correlation between the two, without the need to bring in selection as an additional explanation (lines 209–213). The conceptual idea furthermore states that a strong environmental selection pressure, on the other hand, would be expected to lead to more ARGs

relative to the fecal marker. In line with some (very fair) comment below, we have also stressed more clearly that we can never exclude the presence of some environmental selection with this approach, as selective effects may be subtle and masked behind the noise. Importantly, however, we can conclude that environmental selection would not be a necessary component to explain the observed patterns, the presence of feces is sufficient.

- **It will of course be different in manufacturing waste polluted sites because the antibiotics enter the environment independently of faeces, which is not the case with municipal waste.**

In fact, many industrial effluents, including the Indian site covered here, also treat some human sewage in addition to the wastewater from the pharmaceutical industries (described in e.g. *Larsson DGJ, de Pedro C, Paxeus N. (2007). Effluent from drug manufactures contains extremely high levels of pharmaceuticals. J Hazard Mater. 148:751*). Showing that environmental ARG abundances are very high while fecal markers are non-detectable or barely detectable provides strong support for environmental selection, in particular when this is observed in those environments that have the highest exposure to antibiotics of all environments. Previous studies of ARGs in environments polluted by waste water from antibiotic manufacturing have not investigated the potential co-occurrence of fecal markers. To make this clearer we have also cited the mentioned publication describing the treatment plant more precisely.

- **There is also the possibility that selection for faecal taxa results in further phage proliferation, further reducing the utility of the approach to disentangle dissemination from selection.**

Reviewer #2 is absolutely right, one cannot exclude such a scenario. But if that was the case, it would generate a quite different pattern than the one we observe with rather stable relations between ARGs and phage. We have tried to clarify this in the introduction.

Introduction:

- **“However, as CL1s can contain a wide array of resistance genes and, thus, can be subjected to selection themselves, they are not an independent measure which can be used to assess the selection or dissemination of ARGs”.**

I'm not sure why the authors focus on the independence of CL1s from ARGs as a negative, it is exactly this linkage which makes them a useful proxy for a wide range of AMR mechanisms.

We apologise if this has been unclear and we have tried to further clarify the importance of the independence of the fecal marker and the ARGs. Class 1 integrons have been proposed as markers for anthropogenic (fecal) impact in several studies. Here, we stress that the fact that CL1 most often contain ARGs or are present in the same mobile elements or bacteria (as the reviewer also acknowledge) render them rather useless in disentangling environmental selection from simple dissemination of fecal matter. To clarify further, increased fecal load would lead to increases in both CL1 and ARGs. Similarly, increased selection pressure from antibiotics in the environment would be expected to increase both CL1 and ARGs. This is in contrast to a fecal marker (phage) that is not genetically linked to ARGs, and therefore its abundance is not expected to increase in response to an environmental antibiotic selection pressure making it useful in detecting an environmental selection pressure for ARGs. In agreement with the reviewer, we still think that CL1s are a good proxy for a wide range of AMR mechanisms and have not tried to question that.

- **“Moreover, it is not physically linked to resistance genes like integrons are and hence likely independent from ARGs and thus, could enable the determination of ARG horizontal gene transfer and selection patterns in receiving environments”.**

The authors need to clarify how this would work as its not clear.

We thank reviewer #2 for pointing out this unclear sentence. We have tried to clarify this by mentioning that crAssphage would not be co-selected with the ARGs and thus could be used as an independent marker for fecal pollution (line 230) and tried to make the concept clearer in the introduction (lines 231-236).

- **“A robust marker for fecal pollution would provide the means for distinguishing between onsite selection and dissemination of the genes versus accumulation and to observe decrease in ARGs which is due to dilution of fecal pollution in the receiving environments”.**

I don't follow this logic, due to the factors mentioned above, ie. co-correlation of faecal load and antibiotic concentration and the potential for further phage proliferation I'm not sure I accept the hypothesis. Because WWTP discharges are continuous and there will be a distance decay / dilution effect for ARGs and antibiotics one would expect there to be a correlation between ARGs and faecal pollution regardless of whether dissemination / or selection was the primary driver. The authors later say there is no association between the phage and ARG abundance in human feces which rings alarm bells.

One would indeed expect a correlation between ARG and fecal pollution if dissemination via feces is the main driver of the the presence of ARGs. If the antibiotics (that come together with fecal matter, ARGs and crAssphage via the sewage) would select for and enrich ARGs, then one would expect a shift towards more ARGs relative to crAssphage (which is not expected to be selected for by antibiotics). So, if environmental antibiotic selection is the primary driver for the presence of ARGs in the environment (as in the case with industrial, high level discharges of antibiotic), then the correlation with the fecal marker would be lost or at least be considerably less obvious. So no, at this point we respectfully do not agree with reviewer #2. We also do not understand why it should ring an alarm bell that there is no association between the phage and ARG abundance in human feces. We think that this is one of the strong points of crAssphage as a fecal marker. If the phage is not present in those bacteria that are main carriers of ARGs, then one would not expect an association. Or in other words, people that happen to carry more ARGs than others do not carry more phage than others.

- **There also appears to be some uncertainty regarding the host of the crAssophage, so there is a possibility that this can replicate in environmental strains of Bacteroides and/or other taxa.**

Yes, reviewer #2 is right, this is possible. However, it doesn't seem to be the case as the abundance of crAssphage decreases with increasing distance and we don't see any increase in crAssphage abundance in downstream environments.

- **Only 6 studies appear to be from human impacted environments, whereas there are 484 from environmental metagenomes (presumably not human impacted) and 308 metagenomes were from the human gut. Presumably the differences found may be a result of this significant disparity in sample size.**

The 6 studies were selected because the data in them was available and well documented, which unfortunately is not the case with many metagenomic studies. Then we also analysed public MG-RAST metagenomes, which were previously analysed (Pal et al, 2018) and still available in our lab. These metagenomes were from various sources, including human impacted environments, as can be seen from the results. The ones where we detected crAssphage are polluted with human fecal material, in other words human impacted. The human metagenomes that we included were only used to investigate the association or lack of association between crAssphage, ARGs and CL1s. We here show the independence of crAssphage, and conversely the dependence, between the abundance of crAssphage and CL1s on ARGs in human gut. We think that these datasets address the hypothesis in this manuscript very well and show clearly in each case that the observed ARG patterns in most human impacted environments can be explained by fecal contamination rather than selection. The set of metagenomes also analysed in Pal et al. (2016) excluded small datasets, and we cannot really see how the existing differences in sizes of the metagenomes could have influenced our conclusions to any major extent.

Methods

- **If there is no correlation between the phage and ARGs in human feces why would it be a suitable marker and demonstrate a strong association with ARGs in effluents and receiving waters if it is just a matter of dilution during dissemination?**

We apologise that we have failed in explaining the concept in detail and have tried to make this clearer. The key concept is that If ARGs and the phage would have correlated in human feces (taken from different people) that would have indicated that e.g. antibiotics use could select for both (directly or indirectly), but that seems not to be the case. Environmental fecal pollution is usually the result of pollution with sewage (pooled feces from many people) hence despite independent variations in both ARG and phage levels between people, sewage generated

from a population would be expected to have more stable levels of ARGs and phage (in certain ratio that very well could vary between populations). Hence environmental samples from the same geographical area and which contain different amounts of fecal residuals from the same population, would be expected to harbor both ARGs and crAssphage in relation to how much feces there is in the sample - if there is no selection that has enriched ARGs. So, indeed, if it is just a matter of dilution during dissemination, one would expect that crAssphage and ARG correlate well. And that is what we see.

Results and discussion

- **“Our results show that the observed ARG abundances strongly correlate with crAssphage, meaning that the fecal pollution levels largely explain the observed abundances and there are now clear signs of wide scale selection or dissemination of antibiotic resistance in the affected environments”.**

Again the meaning is not clear hear, do the authors mean “no clear signs”? There is a growing body of literature demonstrating increased prevalence of ARGs in natural environments impacted with WWTP effluent / sewage sludge, so again I don’t know what the authors mean or how they can conclude this from the study of a very small number of human impacted sample sites?

Yes, we mean “no clear signs”. We apologize for the typing error which is now corrected. About the second point, it is clear that WWTPs introduce ARGs to downstream environments and we are by no means trying to deny this, but rather reinforcing it. The point we are trying to make is that the observed ARG patterns are the consequence of fecal pollution and that there are no indications of a wide scale selection or further propagation of the ARGs is detected in the downstream environments (with the exception of industrially polluted sites). The ARGs and crAssphage would not correlate so strongly if there would be a large selection pressure for the ARGs, as can be seen in the case of sediments contaminated with very high levels of antibiotics from manufacturing discharges.

- **“We did not find any correlation between the ARG and crAssphage abundance in fecal metagenomes of the studied populations (linear regression, $F = 25.51$, adj. $R^2 = 0.21$, $p > 0.05$, Figure 1, Suppl. Table 2), confirming that ARG abundance is independent from crAssphage abundance. On the other hand, the correlation between total ARG abundance and int1 gene abundance was significant when taking in to account the different base levels of ARG abundance in the populations (linear regression, $F = 20.55$, adj. $R^2 = 0.29$, $p < 0.05$, Figure 1, Suppl Table 2) showing the expected dependence between ARGs and CL1s”.**

Surely this undermines the suggestion in the introduction that 1) the phage is a good indicator of fecal pollution and 2) that CL1s are not good markers of AMR?!

We again apologise if the concept has not been explained in enough detail in the introduction and have tried to clarify this in the revised manuscript. The lack of correlation makes this approach possible since a selection for ARGs won't affect the crAssphage abundance. ARGs and crAssphage co-exist in human feces but are not linked. So, the observed abundances are independent from each other. Also, we do not claim that CL1s are not good markers for AMR, in fact we confirm that they are. What we claim is that precisely because CL1s and ARGs are strongly linked, analyses of CL1s does not provide clues to if ARGs in certain environments are elevated as a consequence of environmental selection or simply because a high load of fecal bacteria - as CL1s would increase similarly in both situations!

- **“In sludge, the ARG abundance seems to be even lower than expected based on the crAssphage abundance (Suppl. Figure 6). These results speak against WWTPs being hotspots with a strong antibiotic resistance selection acting as a driver for horizontal gene transfer (Guo et al., 2017; Rizzo et al., 2013) and show that at least in the studied treatment plants the treatment process eliminates ARGs along with fecal pollution with high efficiency”.**

Perhaps because crAssphage are replicating in WWTPs or because the majority of sludge is made up of bacteria that replicate within WWTPs.

We appreciate the concern from reviewer #2 and think that the first proposed explanation refers to that a lower than expected ratio of ARG/crAssphage is not necessarily the result of a reduction of ARGs but could alternatively be explained by an increased concentration of crAssphage. This is one possible scenario. However, it can be seen from Suppl. Fig. 6 that there is a 10-fold decrease in crAssphage abundance from influent to

sludge. In light of this observation, we don't think that crAssphage is replicating in the sludge to any major extent. We believe that it is more probable that crAssphage's host bacteria settle in the sludge and are therefore more abundant in sludge than in effluents.

With regards to the reviewers second alternative explanation (that sludge is largely made up of bacteria replicating in the WWTPs), we are less sure what he/she refers to, but we think the idea is that relative proportion of fecal bacteria would be smaller if this is the case - and thereby also the abundance of ARGs and crAssphage reduced. We of course cannot exclude that the fecal bacteria that harbor crAssphage are for some reason reduced more than the taxa which carry more ARGs. If there are differences in the reduction of bacteria carrying ARGs or crAssphage in the sludge, one could argue that it would require a stronger, direct selection pressure for the ARG-carrying bacteria in order to change the ARG/crAssphage ration towards more ARGs. Or to phrase it more generally, differential survival and growth of bacterial taxa that tend to carry ARGs or crAssphage would add to the noise, and thereby reduce possibilities of detecting small effects of environmental selection of ARGs. We have now added this sentence to the revised manuscript (Lines 935–939).

- **“It should be noted that selection occurring on small scale, which does not affect the entire population, or selection of limited types of resistance genes would probably not be detected using this method. Furthermore, rare horizontal transfer events will definitely be missed and more precise methods are needed”.**

This is probably what actually happens, if all ARGs increased significantly in prevalence in all taxa selection would be easy to detect.

We thank reviewer #2 for pointing this out. In the revised version, we have elaborated more on this possibility in line with the reviewer's concern. We have now stressed that the relative proportion of individual ARGs have been reported to be increase in e.g. certain environments exposed to sewage/low levels of antibiotics, which could be an indication of small scale selection. However, community-based analyses such as qPCR or metagenomics does not provide evidence for within-species selection, and such increases of individual ARGs would hence more easily be explained by simple changes in species composition, with no direct link to the exposure to the antibiotics. Furthermore, the lack of consistency on the type of antibiotics the presumably selected for ARGs provide resistance to provides further support to the explanation that the increases are not due to a direct antibiotic selection pressure.

In summary I don't think the manuscript is well argued and there are several logical problems with the argument and conclusions. There are also methodological issues based on the knowledge of the ecology of the specific phage and the small number of impacted metagenomes studied.

We hope that the revised version of the manuscript would be better argued and the concept clearer. We don't think that there are logical problems in our arguments and conclusions. We understand from the comments received that critical parts of the concept has not been understood by the reviewer. We have therefore, where appropriate, tried to clarify the argumentation and logics in the revised manuscript. We also hope that our responses in this rebuttal letter have helped.

Reviewer #3 (Remarks to the Author):

This manuscript is of considerable interest as it concludes that the presence of antibiotic resistance genes in natural environments broadly correlates with the presence of human faecal pollution. If correct, this conclusion will have an important impact on our understanding of the spread of antibiotic resistance genes across the planet and the role of human activity in this process.

We thank reviewer #3 for acknowledging the broad importance and interest in our study

The authors use as a marker for human faeces the recently discovered crAssphage which is claimed to be specific to human faeces. The specificity of this marker is cause for some concern: Garcia-Aljaro et al 2017 conclude that 'crAssphage was also found in samples contaminated with faecal remains from certain animals, though in lower concentrations [approximately 100-fold] than in the human sources' so it is not uniquely specific to humans. The authors have addressed this by analysing the abundance of another Bacteroides phage Φ B124-14, which is found in both humans and animals. However, this

analysis is not shown, even though it is crucially important. It needs to be included and discussed in more detail in a revised version of this manuscript. As Suppl Fig 4 shows high levels of resistance genes can be present in samples from environments that are (presumably) contaminated with animal faeces. It would be very important to know the abundance of Φ B124-14 as a human + animal faecal marker in these samples. It appears that the authors may be overly focused on crAssphage and thus are too anthropocentric in their focus on human faecal pollution as animal faeces appear to be another major source of ARGs in the environment.

Reviewer #3 has a very relevant point in here and we would ideally like to be less anthropocentric. However, this was not possible as good markers for fecal pollution from other animals have not been found yet. We would argue that similar markers for different animal species should be developed and published in a separate paper. The lack of such markers forced our focus to be on human fecal contamination.

We did analyses using the phage Φ B124-14, but found that it performed poorly. We have now included all the data on the phage Φ B124-14 as requested by the reviewer (supplementary table 3) and added some discussion, although not lengthy, as Φ B124-14 phage does not work as a marker for animal fecal contamination and performs even worse as a marker for human fecal contamination (see lines 378–383). Hence, to date to our best knowledge, there is no good marker discovered for animal fecal contamination that would provide similar opportunities for retrospective analyses of shotgun metagenomes as crAssphage provides for human fecal contamination.

Reviewer #3 is right – that animal feces can be another major source of ARGs in the environment. This is probably the case in the samples with high ARGs levels in Suppl Fig 4, as we have also pointed in the manuscript. The MG-RAST annotations also support this, as the samples with high ARG abundances and no crAssphage are from animal sources (e.g. animal habitation, cow shed). Given that the animal species we analysed did not carry crAssphage and that we consistently find correlations between crAssphage and ARGs, it is probable that human fecal contamination is the main source of ARGs in those environments and that the low concentrations of the phage in animal feces did not matter for analyses. Had we found and analysed datasets from environments with very little crAssphage and high abundances of ARGs, contamination with animal feces could have been an alternative explanation, beside environmental selection of ARGs. We have now pointed out the role of animal feces as an important additional source of ARGs and the restrictions of using a marker only specific to human feces (lines 386–388) in the revised manuscript.

One major methodological issue is that there may be a real risk for spurious hits with a minimum match length of only 20 amino acids. Would it not be better to use as a cut-off a match length based on the length of the gene/protein (e.g. >90% of full-length gene/protein) and would this analysis importantly change the outcomes of the analyses? The somewhat unexpected observation that the abundance of *oqxA* and *oqxB* is poorly correlated, even though these genes are generally co-localised, may potentially be explained by 'hitting' a remotely related gene to either *oqxA* or *oqxB* that is identical over a stretch of 20 amino acids. I would like to see some analyses with a more stringent cut-off to determine whether this could really be an issue.

We agree that there is a possibility of false positives and we probably have some. However, as the dataset available are short metagenomic reads (~100–150 bp), it is not possible to target the whole gene. We think that the cut-off used is a good compromise as it requires at least 60 bp to align to the ARGs. And we think that the results speak in favour of the methods, as we would not have so clear results, if we would have a lot of noise in the results. We were the first to use shotgun metagenomics to analyse ARGs in metagenomes (*Kristiansson E, Fick J, Janzon A, Grabic R, Rutgersson C, Weijdegård B, Söderström H, Larsson DGJ. (2011) Pyrosequencing of antibiotic-contaminated river sediments reveals high levels of resistance and gene transfer elements. PLoS ONE 6:e17038. doi: 10.1371/journal.pone.0017038. (Highlight in Nature 2011. doi:10.1038/news.2011.46).*), and we have refined the methodology over the past seven years. We recently published a methodology paper on best practises for analyses of ARGs in shotgun metagenomics datasets (*Bengtsson-Palme J, Larsson DGJ, Kristiansson E. (2017). Using metagenomics to investigate human and environmental resistomes. J Antimicrob Chemother. 72:2690*). We have followed this practise also in the present study.

Minor issues:

- Please add page and line numbers. For any comment below I have simply copied the relevant line, which you should be able to find in the document using the 'Find' function.

The revised version includes line numbers.

- There a small number of minor issues with language or style in the manuscript. I have not identified all of them here.

We have tried to correct all language issues throughout the whole manuscript.

Abstract

- 'This was possible by analyzing the abundance of a newly discovered phage': please name the phage in the abstract

Corrected as suggested.

Introduction

- 'Fecal pollution levels have rarely been incorporated in the determination of possible section': replace section by selection.

Corrected as suggested.

- The authors may want to add a reference to the recently published study by Stachler et al (Environ. Sci. Technol., 2018, 52 (13), pp 7505–7512) to the manuscript.

We have added the reference. It was apparently published one day before we submitted our manuscript, explaining why we missed it.

Materials and methods

- I initially read '12 chicken (Xiong et al., 2018), a subset of 100 samples from a bigger pig metagenome study (Xiao et al., 2016)' as meaning that the 12 chicken samples were extracted from a larger pig metagenome study that was not further analysed. Perhaps this needs to be rephrased for clarity.

We have rephrased the sentence.

- It is unclear what the authors mean with 'longest ORF was selected for each gene'. As ResFinder holds unique genes it is not entirely clear what the authors have done. E.g. have they clustered genes to a certain threshold of nucleotide conservation and then selected as a representative gene for the cluster the longest gene in each 'gene cluster'?

Unfortunately ResFinder database has some inaccuracies and sometimes the resistance gene sequence contains more than just the ORF for the gene. So to be able to translate the ResFinder database to amino acid sequences automatically, this approach was used.

Results and discussion

- 'and there are now clear signs of wide scale': this is a crucial typo as I believe the authors mean 'no clear signs'

Corrected as suggested.

- More information on the types of resistance genes should be provided, e.g. by providing a supplementary dataset that lists the abundance of each individual resistance gene. This will reveal whether there are specific antibiotic resistance genes that are highly abundant and that mostly determine the 'normalized ARG abundance' as used in the main text.

We have provided the abundance of individual genes for the selected studies in Suppl. Table 4. We excluded the MG-RAST and WWTP samples as they have been analysed for ARGs in the original publications. From these publications it can be seen that the observed ARG/crAssphage patterns observed in this manuscript are not determined by few very abundant ARGs.

- **'Using proxies for fecal pollution such as crAssphage enable detecting selection and thus, can help in assessing risks associated with it.'** This is clearly wrong: crAssphage may be used to detect the presence of human faeces and the resistant bacteria that are associated with it, but it does not detect selection for antibiotic resistance in an environment.

Reviewer #3 is right, it does not. It helps interpreting causes, where a stable ARG/crAssphage ratio would suggest that dissemination of human fecal bacteria is the main explanation for the levels of ARGs found, whereas as high ARG/crAssphage would suggest that additional mechanisms are involved, such as environmental selection of ARGs and/or animal fecal contamination with bacteria carrying ARGs. We now modified this statement accordingly.

Supplementary data

Supplementary tables 1 and 2 need informative legends.

We have added informative legends to suppl table 2. Suppl table 1 has only accession numbers and the dataset is indicated in table name.

Figures

Is the y-axis in Fig 2 – 4 correct? E.g the normalised ARG abundance cannot be 10^{1000} but should probably be 10^3

We apologize for the confusion. The "log10" on the axis label is misleading. We have now corrected it.

REVIEWERS' COMMENTS:

Reviewer #2 (Remarks to the Author):

Having read the responses and revised paper I still think there are too many assumptions in this manuscript and the interpretation of the results is too speculative. With only 6 studies in the test data set and from what I can see no accompanying antibiotic residue data this analysis is too preliminary. In addition the inclusion of the Indian antibiotic production sample, which is exceptionally high in antibiotics and low in faecal pollution, reduces the apparent differences between the other sample sets in the multivariate analysis. Without that data much greater differences would be apparent between the other samples. The Indian antibiotic polluted site is in effect an outlier. With the uncertainties around phage replication in situ and the previous points, I don't see how this really disentangles dissemination from selection or how it improves understanding from previous studies showing that there is an association between faecal pollution and ARG prevalence.

Reviewer #3 (Remarks to the Author):

The authors have addressed practically all of the reviewers' comments in the revised version of their manuscript. The points below may still deserve more attention.

I. 145 - 147: this line is poorly structured and needs to be rewritten. More importantly, I do not see how the approach in this study can provide information on horizontal gene transfer as linkage of antibiotic resistance genes to microbial hosts is not studied.

I. 191 – 192: while I understand the authors having to filter hits in the Resfinder database, it is still not clear to me why they have selected the longest hit for their analyses. A rationale needs to be provided in the text of the manuscript.

I. 198: the rationale for choosing the cut-off of an alignment at 20 amino acids is unconvincing in the 'response to referees' letter. On the one hand the authors admit that they likely have false positives, but that they are limited by the sequence read length (100 – 150 nucleotides). The authors appear to be unwilling to expand their cut-off (e.g. 30 amino acids), so I would suggest they add a comment that the cut-offs are not extremely stringent (and may result in false positives) but are supported by previous publications, as indicated in the response to the referees.

REVIEWERS' COMMENTS:

Reviewer #2 (Remarks to the Author):

Having read the responses and revised paper I still think there are too many assumptions in this manuscript and the interpretation of the results is too speculative. With only 6 studies in the test data set and from what I can see no accompanying antibiotic residue data this analysis is too preliminary.

Figure 2 is indeed based on six studies (with together more than 50 different metagenomes). Including the other figures, we based the overall analysis on nearly 500 metagenomic datasets. Nevertheless, we agree that more data would provide deeper insights into the dynamics (see end of discussion) but argue that the overall picture is convincing already with the data at hand.

With regards to the lack of accompanying antibiotic residue data, It would indeed be useful to have also the antibiotic concentrations from each study. Importantly, however, that would likely not change our analyses, results or interpretation. Regardless of the antibiotic concentration present in the studied environments, our analyses would still reveal a clear correlation between ARGs and faecal pollution and decreasing concentrations with increasing distance from the pollution source in the analysed datasets (with the exception of the Indian one).

From the Indian samples we know that they have very high antibiotic concentrations, even above therapeutic levels. Hence, this strongly support our finding of a lack of correlation between ARGs and crAssphage at these sites (as a reflection of the selection of ARGs). Whereas most studies included here have not quantified antibiotic concentrations, we can conclude, based on a rather large literature of antibiotic residues in sewage and sewage-impacted environments, that the levels found are always much, much lower than from industrially polluted sites (such as the study sites in India). Antibiotic levels were indeed available for the Swedish WWTP samples, and they were, as expected, far below known inhibitory concentrations for bacterial growth. Hence, it would make perfect sense that these concentrations provide no or a very small selection pressure, in line with our results.

In addition the inclusion of the Indian antibiotic production sample, which is exceptionally high in antibiotics and low in faecal pollution, reduces the apparent differences between the other sample sets in the multivariate analysis. Without that data much greater differences would be apparent between the other samples. The Indian antibiotic polluted site is in effect an outlier.

We have not done any multivariate analyses as reviewer #2 implies, we have only performed several, simple univariate regressions. The Indian samples were, as the other datasets, analysed in a separate regression model. Hence, removing the Indian samples would not change the results in any way. We agree that the Indian samples could indeed be called "outliers" since they were the only samples where we could not explain the antibiotic resistance gene patterns with faecal pollution. However, detecting these 'outliers' is one of the reasons why we think that using crAssphage is so useful. It enables us to find environments where ARGs are not correlated with fecal pollution suggesting that there is some other force driving the enrichment that calls for further studies.

With the uncertainties around phage replication in situ and the previous points, I don't see how this really disentangles dissemination from selection or how it improves understanding from previous studies showing that there is an association between faecal pollution and ARG prevalence.

We agree that it would be valuable to understand if there is a potential for crAssphage replication in situ which we also have indicated in the manuscript. Importantly, however, if there were a significant level of on-site replication of crAssphage in the environment, it would likely contribute to a less good correlation between crAssphage and ARGs. Please note that that we (still) find a good correlation, which strongly supports our conclusion that ARG levels largely can be explained by fecal pollution levels. We are also not aware of any other study that have attempted to correlate fecal markers in metagenomes with ARGs on a large scale, hence we do think the present study provides an important contribution to our understanding of the association between fecal pollution and ARG prevalence.

Reviewer #3 (Remarks to the Author):

The authors have addressed practically all of the reviewers' comments in the revised version of their manuscript. The points below may still deserve more attention.

We are happy to see that reviewer #3 appreciated how we had addressed the great majority of his/her concerns.

I. 145 - 147: this line is poorly structured and needs to be rewritten. More importantly, I do not see how the approach in this study can provide information on horizontal gene transfer as linkage of antibiotic resistance genes to microbial hosts is not studied.

We have rewritten the sentence and removed claim on HGT

I. 191 – 192: while I understand the authors having to filter hits in the Resfinder database, it is still not clear to me why they have selected the longest hit for their analyses. A rationale needs to be provided in the text of the manuscript.

We have tried to clarify this in the methods section. This was only used in the translation step. We needed amino acid sequences to be able to use Resfinder with DIAMOND in ARG annotations. The actual annotations have not been done based on longest hit.

I. 198: the rationale for choosing the cut-off of an alignment at 20 amino acids is unconvincing in the 'response to referees' letter. On the one hand the authors admit that they likely have false positives, but that they are limited by the sequence read length (100 – 150 nucleotides). The authors appear to be unwilling to expand their cut-off (e.g. 30 amino acids), so I would suggest they add a comment that the cut-offs are not extremely stringent (and may result in false positives) but are supported by previous publications, as indicated in the response to the referees.

We have acknowledged this in the revised manuscript.